# Laboratory Liquid-Jet X-ray Microscopy and X-ray Fluorescence Imaging for Biomedical Applications

**DOI:** 10.3390/ijms25020920

**Published:** 2024-01-11

**Authors:** Komang G. Y. Arsana, Giovanni M. Saladino, Bertha Brodin, Muhammet S. Toprak, Hans M. Hertz

**Affiliations:** Department of Applied Physics, Biomedical and X-ray Physics, KTH Royal Institute of Technology, 10691 Stockholm, Swedensaladino@kth.se (G.M.S.);

**Keywords:** liquid-jet X-ray source, X-ray fluorescence imaging, X-ray microscopy, cell imaging, bioimaging, multiplexed imaging, stain-free imaging, nanomedicine

## Abstract

Diffraction-limited resolution and low penetration depth are fundamental constraints in optical microscopy and in vivo imaging. Recently, liquid-jet X-ray technology has enabled the generation of X-rays with high-power intensities in laboratory settings. By allowing the observation of cellular processes in their natural state, liquid-jet soft X-ray microscopy (SXM) can provide morphological information on living cells without staining. Furthermore, X-ray fluorescence imaging (XFI) permits the tracking of contrast agents in vivo with high elemental specificity, going beyond attenuation contrast. In this study, we established a methodology to investigate nanoparticle (NP) interactions in vitro and in vivo, solely based on X-ray imaging. We employed soft (0.5 keV) and hard (24 keV) X-rays for cellular studies and preclinical evaluations, respectively. Our results demonstrated the possibility of localizing NPs in the intracellular environment via SXM and evaluating their biodistribution with in vivo multiplexed XFI. We envisage that laboratory liquid-jet X-ray technology will significantly contribute to advancing our understanding of biological systems in the field of nanomedical research.

## 1. Introduction

X-ray has emerged as an important method for biological studies due to its high spatial resolution, high penetration depth, and non-invasive nature on biological systems [1,2,3]. This holds true for both soft X-ray microscopy (SXM) and X-ray fluorescence imaging (XFI).

SXM provides a powerful tool for cellular biology with high resolution and high contrast under unperturbed conditions [4,5,6,7]. The unique properties of soft X-rays, such as the short wavelengths [8,9], allow nanometer-scale resolution of thick samples (>1 µm), making them an attractive alternative to other microscopy techniques for exploring the cell’s interior in 2D or 3D imaging [10,11,12,13]. Additionally, SXM eliminates the need for chemical fixation, sectioning, heavy-metal staining, and optical fluorescence labeling [4,14,15,16].

XFI allows for the non-invasive detection of contrast agents constituted of specific elements, based on their characteristic X-ray emissions [17]. This spectral specificity constitutes a valuable property for analyzing biodistribution and concentration in preclinical studies [18,19,20,21,22,23,24,25]. Contrary to positron emission tomography (PET) and single photon emission computed tomography (SPECT), the possibility of tracking non-decaying markers permits longitudinal studies of the same subjects over long timespans [22]. Furthermore, several elements can simultaneously be tracked for multiplexed imaging [24,26].

Both systems typically operate at synchrotron facilities because they provide high spectral brightness [27,28,29]. These techniques have successfully demonstrated a range of different imaging applications at the cellular level, imaging tissues, and small animals, such as visualizing membranes, determining the shape and size of organelles, detecting nanoparticles (NPs) inside the cell [30,31,32,33], tracking nutrient distribution in plant tissues [34], in vivo tracking of immune cells [35], and assessing the toxic effects of NPs in small animals [36]. However, a limited number of synchrotron facilities results in limited access to these methods by biological scientists. Laboratory-based X-ray sources are being developed as a complement to synchrotron-based instruments with comparable capabilities.

The X-ray source is an important aspect of X-ray imaging techniques: it has to deliver sufficient spectral brightness to achieve high contrast and short exposure times. In the laboratory setup, this can be achieved by using a liquid-jet X-ray source. In the soft X-ray regime, the liquid jet target is proven to produce more sustainable photon flux than solid bulk, solid tape, and gas-puff targets [7]. Similarly, in the hard X-ray region (24–29 keV), the maximum e-beam power density of the liquid metal jet source is 100× higher than conventional tungsten-solid-anode microfocus X-ray source [37], enabling the tracking of contrast agents in vivo with high sensitivity, specificity, and spatial resolution through XFI [25,38,39].

In the present work, we developed a methodology to study NP uptake and interaction in vitro and in vivo using liquid-jet X-ray sources. We employed a liquid nitrogen (LN) jet laser plasma source that operates in the water window region (0.5 keV) for cellular studies and a liquid–metal jet microfocus source that operates in the hard X-ray region (24 keV) for preclinical imaging of small animals. This study demonstrates the capability of liquid-jet X-ray sources to observe NPs inside the cell and evaluate their biodistribution in vivo. Our findings complement previous studies conducted with other imaging techniques and propose the liquid-jet X-ray imaging technique as a robust tool to investigate bio-nano interactions both in vitro and in vivo. SXM and XFI enabled a synergistic approach that could advance the understanding of NP interactions at both cellular and whole-body levels.

## 2. Results and Discussion

### 2.1. Liquid-Jet X-ray Sources

The main components of our laboratory liquid-jet laser-plasma SXM setup are schematically shown in Figure 1a [8]. The SXM liquid jet is formed by cooling high-purity gaseous nitrogen to −192 °C, transforming it into liquid nitrogen. This liquid form of nitrogen is then driven through a 25–30 µm fused silica capillary nozzle by applying a pressure of 8–20 bars. The nitrogen plasma is generated by focusing on a 1064 nm diode-pumped Nd:YAG slab laser with ~500 ps and ~100 mJ pulses at a repetition rate of 2.0 kHz. Normal-incidence multilayer Cr/V condenser is used to collect the λ ≈ 2.48 nm line emission from the laser plasma, which is then focused on the samples [7]. The soft X-ray (λ ≈ 2.48 nm) required a high vacuum system to prevent absorption by air. The operational pressure of 10^−5^–10^−3^ mbar was used for the setup. Cryo-fixation was demonstrated to be the most effective method for preparing samples for liquid-jet laser-plasma SXM [10,14]. The cryo-fixated samples are mounted onto a cryo-sample holder placed on a modified transmission electron microscopy (TEM) sample stage. This setup is particularly suitable for adherent cells. A 30 nm outer-zone-width Fresnel optic finally images the transmitted X-rays onto a CCD detector [40]. 

A metal liquid-jet X-ray electron-impact source is instead used to generate hard X-rays for XFI and X-ray fluorescence (XRF) computed tomography (Figure 1b). The liquid jet as the anode consists of Galinstan, a liquid metal alloy of Ga, In, and Sn. The liquid state of the metal allows us to obtain a 100× increase in the e-beam power density compared to the rotating anode design [37]. By tuning the reflectivity of the focusing optics (multilayer mirror), we exploited the In K_α_ characteristic emission line (≈24 keV) employed as a pencil beam (100 µm spot size) for XFI. This energy matches the absorption edges of several transition metal elements, especially Mo, Ru, and Rh [25]. Furthermore, we previously identified yttrium (Y), zirconium (Zr), niobium (Nb), and rhodium (Rh) as other potential XRF contrast elements, validated with small animal equivalent phantoms and theoretical models [41]. Upon excitation, these elements emit isotropic K_α_ fluorescence emission. The XRF detectors (SDDs) were mounted on the horizontal stage forming an angle of 90° with the pencil beam. A stage allows the translation in the x–y direction and rotation to enable tomographic imaging (XFCT) [42]. In vivo, XFI of small animals (mice) was made possible with the introduction of equipment for anesthesia and respiration monitoring [22]. Purposely designed NPs containing Mo or Ru enabled the performance of longitudinal studies on NP pharmacokinetics by direct detection of their K_α_ emission within the body [22,23]. 

A summary of the information on our liquid-jet-based SXM and XFI is provided in Table 1, detailing the main differences between the two setups. These enable comprehensive biomedical investigation of in vitro cellular studies and in vivo preclinical imaging.

### 2.2. Nanoparticle Design

MoO_2_ NPs and Ru NPs were synthesized using solvothermal and polyol methods, respectively. MoO_2_ NPs exhibited a clustered dry size of 40 ± 12 nm (Figure 2a), while Ru NPs had a quasi-spherical shape with a dry size of 3 ± 1 nm (Figure 2b), estimated using transmission electron microscopy (TEM). Colloidal (hydrodynamic) size distribution was measured for both the NPs with dynamic light scattering (DLS), resulting in an average size of 59 ± 18 nm and 12 ± 5 nm for MoO_2_ NPs and Ru NPs, respectively (Figure 2c). The ζ-potential values for MoO_2_ NPs and Ru NPs were equal to −39 ± 1 mV and 0 ± 1 mV, respectively. The difference between the dry (TEM) size and wet (DLS) size was ascribed to the chemisorbed capping agent (PVP) and adsorbed water on the NP surface. The polydispersity index (PDI) values of 0.11 and 0.04 for MoO_2_ and Ru NPs, respectively, indicated good colloidal dispersibility and stability of the synthesized NPs. The crystalline phases were previously investigated with powder X-ray diffraction (PXRD), highlighting a single dominating phase for both the NPs, corresponding to the hexagonal MoO_2_ (ICDD card No.: 00-050-0739) and metallic Ru (ICDD card No.: 01-089-4903) [26]. In the previous study, both MoO_2_ NPs and Ru NPs were tested as potential candidates for XFI using real-time cell analysis assay for cytotoxicity evaluations [26]. 

### 2.3. In Vitro X-ray Microscopy Imaging

In the methodology proposed here, X-ray micrographs of RAW 264.7 macrophages were acquired using our laboratory liquid-jet laser-plasma SXM (Figure 1a). The cells exposed to MoO_2_ NPs were seeded onto gold TEM grids (Figure 3a) for 17 h to adhere and permit phagocytosis. RAW 264.7 macrophages were chosen because of their predominant role in the immune response to intravenously injected nanomaterials [43,44]. High-resolution imaging of cellular structures and NPs is attainable with cryo-fixated samples without any staining or labeling. This is possible due to the utilization of a 0.5 keV soft X-ray beam (Figure 3b). The source energy (0.5 keV) is located in a region between the carbon K-edge (λ = 4.3 nm) and oxygen K-edge (λ = 2.3 nm), called a water window [7,9]. In this energy range, the organelles and membranes (mostly consisting of carbon and nitrogen) absorb the majority of incoming X-rays, whereas the cytoplasm permits significant X-ray transmission owing to its higher water content. Moreover, MoO_2_ NPs exhibit the highest absorbance ascribed to the high atomic number of the metallic element (Z_Mo_ = 42). 

SXM has a significantly higher penetration depth in comparison to other high-resolution imaging techniques, enabling the imaging of fully hydrated cells with thicknesses of up to ~10 µm [4,7,9]. SXM micrographs of untreated control (Figure 4a,b) and NP-exposed (Figure 4c) macrophages were acquired and compared. The selected cell line turned out to be optimal for SXM due to its thickness and its ability to readily attach and spread on the holey carbon grid used in this experiment (Figure 4a). The distinctive globular shape observed in Figure 4b, preceding cell detachment, allows for easy differentiation of dying cells from healthy cells.

Nuclear membrane and nucleolus can be easily identified due to the high natural contrast of SXM on organelles with high carbon density, despite the absence of any staining (Figure 4a,b). The SXM also offers a resolution of a 25 nm half period [45], making it possible to observe the nuclear membrane (Figure 4b). NP accumulations can be observed within the cytoplasm 17 h after NP exposure due to the macrophage phagocytic process, as presented in Figure 4c. The intrinsic elemental contrast of heavy metals (e.g., Mo) permitted localization of MoO_2_ NPs retained in quasi-spherical organelles (up to a few microns in diameter) in the cytoplasm of cells, ascribed to the formation of phagosomes and lysosomes in the phagocytic pathway [46]. These constitute the cellular digestive system involved in the elimination of waste materials from phagocytic activity [43]. These observations validated previous observations with sectioned slices of cells exposed to MoO_2_ NPs and imaged using electron microscopy [26]. 

SXM overcomes the need for optical fluorescence staining for cell studies, usually achieved by conjugation to the NP surface or by coating with a dye-doped shell [26,47,48]. SXM also offers high-resolution visualization capabilities for single-cell imaging. In the future, SXM could be employed to study alterations in cell and bacterial membranes following NP-based therapy, commonly investigated using scanning electron microscopy on metal-sputtered samples [49]. Furthermore, longitudinal studies with SXM might shed light on cellular mechanisms and interactions with NPs in a near-native stage, eliminating the need for resin embedding and sectioning for TEM imaging [26,50]. Cell imaging at a near-native state could ease the evaluation and subsequent selection of biocompatible NPs for diagnostic and therapeutic application within the nanomedicine field. 

### 2.4. In Vivo X-ray Fluorescence Imaging

Contrast-mediated XFI was enabled by using elements whose electron absorption edge matches the X-ray source energy [38,42,51]. The energy-resolved signal acquisition permits the simultaneous detection and localization of several elements, such as Mo, Ru, and Rh, through their K_α_ XRF emission [26]. The XRF signal was collected by a photon-counting silicon-drift detector (SDD, RaySpec Ltd., England, UK) positioned at a 90° angle relative to the pencil beam. Simultaneously, the transmission signal passing through the sample was measured using another photon-counting SDD (Figure 1b). 

Using diluted samples from the NP stocks, we employed the metal liquid jet setup to estimate NP concentration (Figure 5a) by obtaining calibration curves with Mo (Figure 5b) and Ru (Figure 5c) standard dilutions within the [0, 100] ppm range. Mo K_α_ and Ru K_α_ fluorescence emission signals were identified at 17.4 keV and 19.2 keV, respectively. The tailing background ascribed to the Compton scattering was removed by subtracting the spectrum acquired from a water-filled vial. The possibility to select the spectral range of interest provides XFI with a characteristic high elemental specificity compared to other clinical imaging techniques such as contrast-enhanced magnetic resonance imaging (MRI) and conventional absorption-based X-ray imaging [39,52]. 

In a previous study, the multiplexing property of XFI/XFCT was demonstrated in situ with a postmortem experiment [26]. In the present work, two mice were co-injected with an equal XFI-active concentration of MoO_2_ NPs and Ru NPs ([Mo] = [Ru] = 20 mg/kg), as schematically depicted in Figure 6a. The mice were anesthetized and, subsequently, imaged 1 h after injection, recording the spectral range for both Mo K_α_ and Ru K_α_ fluorescence emissions. XRF projection images were extracted from the XRF detectors by selecting a spectral range corresponding to the elements of interest, [17.1, 17.7] keV for Mo and [18.9, 19.6] keV for Ru. The current setup could achieve a spatial resolution of 200 µm [12].

MoO_2_ NPs and Ru NPs exhibited a distinct biodistribution (Figure 6b). MoO_2_ NPs predominantly accumulated in the lungs, liver, and spleen, with occasional signals from the bladder. Ru NPs exhibited a scattered biodistribution, indicating a probable longer circulation time compared to MoO_2_ NPs. These findings align with previous in vivo experiments, with mice injected with a single NP kind [22,23]. The co-injection of MoO_2_ NPs and Ru NPs did not influence their macroscopic biodistribution. Thus, the possibility of performing in vivo multiplexed XFI can lead to a reduction in the number of animal experiments in the future and permit the evaluation of combination therapy with ad hoc pharmacokinetic studies. 

## 3. Materials and Methods

### 3.1. Materials

Ammonium heptamolybdate (AHM, (NH4)6Mo7O24·4H2O), ruthenium (III) chloride hydrate (RuCl_3_, Ru 38–40%), poly(vinyl-pyrrolidone) (PVP, 55 kDa), and ethylene glycol (EG, >99%) were all purchased from Sigma Aldrich (Stockholm, Sweden). Ethanol, absolute (EtOH, ≥99.8%) was obtained from VWR (Stockholm, Sweden). A MilliQ reference water purification system (Merck Millipore, Burlington, MA, USA) was used for deionized (DI) water. 

### 3.2. Nanoparticle Synthesis

MoO_2_ NPs were synthesized using a solvothermal method [22,26]. The precursor AHM (3.6 mM) was dissolved in 54 mL of DI water and 24 mL of EtOH. PVP (0.29 mM) was added and stirred for 30 min. The synthesis was performed at 180 °C for 18 h using a stainless steel autoclave with Teflon lining. The obtained MoO_2_ NPs were washed by centrifugation and redispersion in DI water. Ru NPs were obtained via polyol synthesis using RuCl_3_ as the precursor [23,26]. RuCl_3_ (0.2 mmol) was dissolved in DI water (500 µL) and transferred to EG (20 mL). PVP (4 mmol) was subsequently added while stirring. The solution was heated to a refluxing temperature and reacted for 2 h. The NPs were collected by precipitation with acetone and centrifuging. The NP stocks were then stored at 4 °C for further use.

### 3.3. Characterization Techniques

The dry particle size and morphology of the designed NPs were studied using transmission electron microscopy (TEM) (JEM-2100F, 200 kV, JEOL, Tokyo, Japan). Dynamic light scattering (DLS, Malvern Nano-ZS90) was used to measure the hydrodynamic size distribution and ζ-potential of the NPs dispersed in DI water (pH 6.5). Inductively coupled plasma optical emission spectroscopy (ICP-OES) (iCAP 6000 series, Thermo Scientific, Waltham, MA, USA) was used for the determination of the elemental composition of the as-synthesized materials prior to in vitro and in vivo experiments. The concentration was confirmed with XRF spectrum acquisitions of the two NP stocks (Figure 5a) by obtaining calibration curves using several dilutions of Mo and Ru standards. The spectra were acquired with scans of 60 s, in triplicates (±2 SD). The Compton background was removed by subtracting the spectrum of a control sample vial filled with distilled water. Linear fits were performed on the integrated signals for both Mo (R^2^ = 0.995) and Ru (R^2^ = 0.999) data points. The stock dispersions for intravenous injections were analyzed using Limulus Amebocyte Lysate (LAL) assay to evaluate the presence of lipopolysaccharides (LPS). The LPS concentration in all the samples measured was below 0.1 EU/mL.

### 3.4. Cell Studies

RAW 264.7 (murine macrophages, 91062702-1VL, Sigma Aldrich, Stockholm, Sweden) were employed for cell studies. The cells were cultured in a flask until a concentration of 0.8 × 10^5^–1 × 10^6^ cells/mL was reached. Subsequently, the cells were exposed to MoO_2_ NPs at a concentration of 500 µg/mL. The NP-exposed cells were seeded on the back side of gold TEM grids. A 300-mesh holey carbon layer (Agar Scientific, Stansted, UK) was used for cell adhesion. After 17 h, the grids were washed three times with PBS. Following this process, the grids were examined under a light microscope to assess the number of cells on the grid wells. Subsequently, the grids were mounted on plunge freezing setup and plunge-frozen in liquid ethane/propane. Before cryo-fixation, the excess medium/PBS was removed by blotting the grids with blotting paper. This process was monitored using a microscope; the liquid was kept below the grid height (approx. 10 μm). After the plunge-freezing process, the samples were stored at −165 °C. The stored samples were taken out and inserted into the SXM setup using a cryo sample holder specifically engineered to maintain a temperature of −165 °C throughout the imaging process. All SXM images were acquired using the laboratory-based X-ray microscope (Figure 1a). Several cells were imaged on the gold TEM grid, and each grid was scanned using an exposure time of 1 s to locate the sample of interest and an exposure time of 10 s to obtain the best focus; the final images were acquired with an exposure time of 30 s. 

### 3.5. Animal Studies

Animal experiments were approved by the regional animal ethics committee of Northern Stockholm, Sweden (ethical permit number 10579-2020). Two five-week-old SCID female mice (CB17SC-sp/sp) were purchased from Taconic Biosciences (Lille Skensved, Denmark) and housed under controlled temperature (21 ± 1 °C) and humidity (55 ± 5%), with light–dark cycle and ad libitum feeding. Mice were injected intravenously with 100 μL NP suspension containing mixed MoO_2_ NPs and Ru NPs ([Mo] = [Ru] = 5 mg/mL) in PBS, corresponding to a dose of 20 mg/kg per metallic element (Figure 6a). XRF projection images were acquired in vivo with the laboratory liquid-jet XFI setup (Figure 1b). Mice were anesthetized during imaging using isoflurane (Abbott, Solna, Sweden). The cornea was protected with ophthalmic ointment (Oculentum simplex, APL, Huddinge, Sweden). Images were acquired 1 h after NP injection with a step size of 200 μm and exposure time of 10 ms per step, resulting in a 15-min scanning time for a whole-body projection image, previously validated for in vivo imaging [22]. Mo K_α_ and Ru K_α_ XRF signals were acquired and overlaid on the transmission projection image, yielding multiplexed images. The background signal was subtracted from all pixels, with a per-pixel background estimate of 1 photon for Mo and 2 photons for Ru spectral energy ranges. 

## 4. Conclusions

With the presented work, we proposed a synergistic approach for NP evaluation through in vitro and in vivo studies using exclusively laboratory X-ray imaging based on liquid-jet technology. SXM permitted the study of NP uptake by cells in a near-native state by exploiting the natural contrast within the soft X-ray water window to acquire stain-free images of macrophages with phagocyted NPs. Subsequently, XFI enabled in vivo multiplexed imaging of purposely designed NPs as the contrast agents, with high elemental specificity. Despite advantages in terms of accessibility, the utilization of laboratory SXM requires improvement in spectral brightness to achieve better contrast. Additionally, laboratory XFI can only be performed using NPs consisting of specific elements, due to the limited availability of liquid-metal jet sources for other energy ranges. In addressing these limitations, future research could focus on investigating bio-nano interactions with 3D SXM and correlative imaging. Moreover, a broader range of NP types could be tested for in vivo XFI. We envision that the here-presented methodology will enhance understanding of NP interactions at the cellular and whole-body levels.

## Figures and Tables

**Figure 1 ijms-25-00920-f001:**
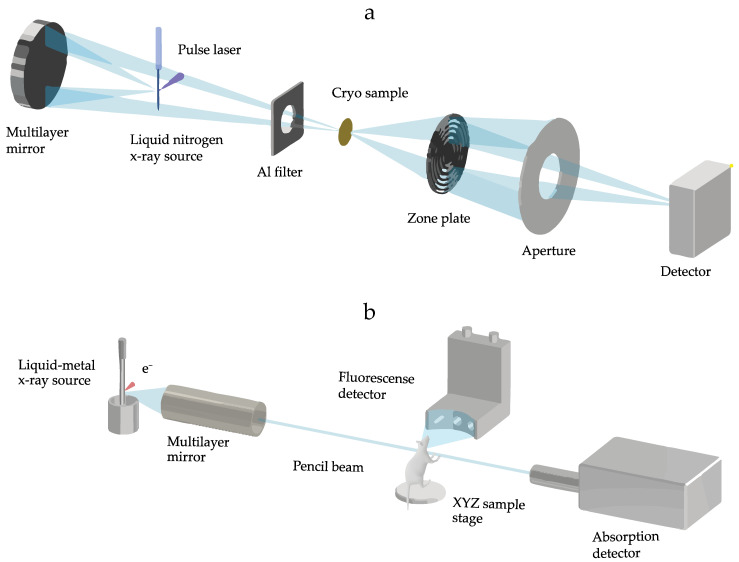
Schematic representation of laboratory liquid-jet (**a**) soft X-ray microscope (SXM) and (**b**) X-ray fluorescence imaging (XFI).

**Figure 2 ijms-25-00920-f002:**
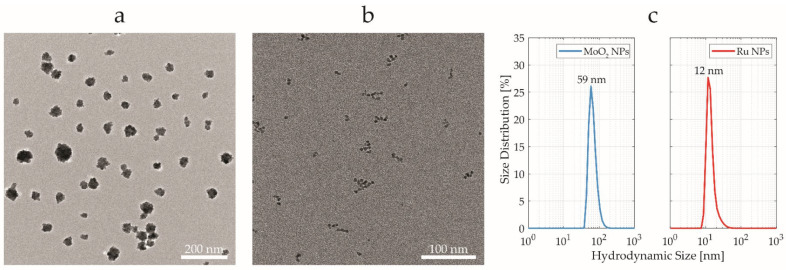
Nanoparticle characterization. Transmission electron microscopy (TEM) micrographs of (**a**) MoO_2_ NPs and (**b**) Ru NPs. Scale bars are (**a**) 200 and (**b**) 100 nm, respectively. (**c**) Hydrodynamic size distribution of MoO_2_ NPs (blue) and Ru NPs (red), obtained by dynamic light scattering (DLS).

**Figure 3 ijms-25-00920-f003:**
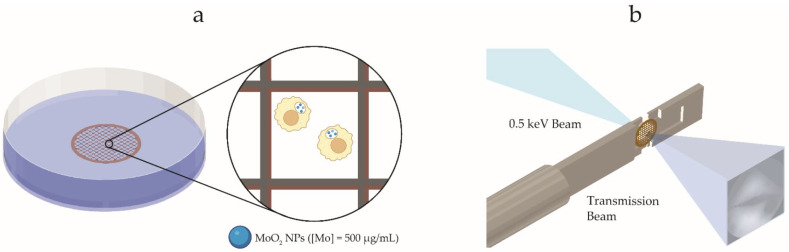
Cell preparation steps for sample imaging with soft X-ray microscopy (SXM). (**a**) Cell adherence on a gold grid and subsequent exposure to MoO_2_ NPs (Mo = 500 µg/mL). (**b**) A soft X-ray beam (0.5 keV) provides natural contrast on unstained samples using the water window.

**Figure 4 ijms-25-00920-f004:**
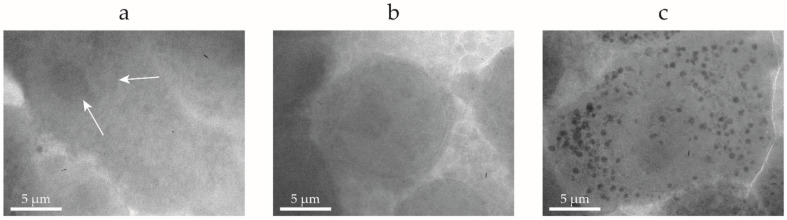
Cell imaging with soft X-ray microscopy. Representative X-ray micrographs of RAW 264.7 macrophages before (**a**,**b**) and after (**c**) exposure to MoO_2_ NPs. The arrows point at the nuclear membrane and nucleolus of a healthy cell (**a**), compared to a dead cell characterized by a globular shape (**b**). Scale bars are 5 µm.

**Figure 5 ijms-25-00920-f005:**
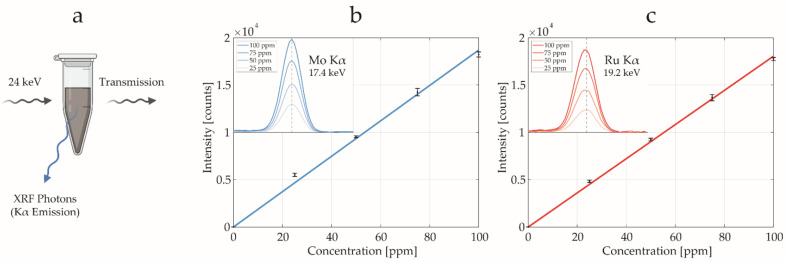
X-ray fluorescence concentration estimation. (**a**) Schematic representation of an NP stock solution scanned with a hard X-ray pencil beam (24 keV). The K_α_ fluorescence emission was recorded to estimate the NP concentration. Calibration curve for fluorescence intensity as a function of the concentration of Mo (**b**) and Ru (**c**) standards. In the insets, the Kα emission peaks are reported as a function of concentration, after background removal.

**Figure 6 ijms-25-00920-f006:**
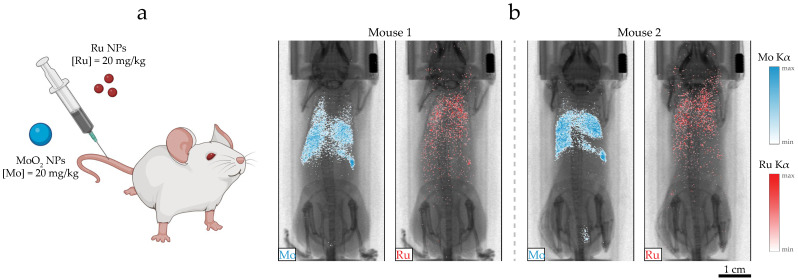
In vivo X-ray fluorescence imaging. (**a**) MoO_2_ NPs and Ru NPs were co-injected intravenously ([Mo] = [Ru] = 20 mg/kg). (**b**) X-ray fluorescence projection images of two mice, 1 h after NP co-injection. Multiplexed signals from Mo K_α_ (blue) and Ru K_α_ (red) highlight the different NP biodistributions. The scale bar is 1 cm.

**Table 1 ijms-25-00920-t001:** Main characteristics of laboratory liquid-jet soft X-ray microscopy (SXM) and X-ray fluorescence imaging (XFI).

	Soft X-ray Microscopy	X-ray Fluorescence Imaging
Source Energy	0.5 keV	24 keV
Imaging Mode	Full Field	Scanning
Liquid-Jet Material	Liquid Nitrogen	Galinstan (Metal Alloy)
Contrast	Photoelectric Absorption	K_α_ Fluorescence Emission
Application	Cellular Imaging	Animal Imaging

## Data Availability

All the data supporting the findings of this study are available within the article. Raw data are available from the corresponding authors upon request.

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
