# Peer review of "Laboratory Liquid-Jet X-ray Microscopy and X-ray Fluorescence Imaging for Biomedical Applications"

_ijms, 2024, doi:10.3390/ijms25020920_

Round 1

Reviewer 1 Report

Comments and Suggestions for Authors

The manuscript under review outlines a novel technique for soft X-ray imaging and X-ray fluorescence imaging, utilizing a laboratory setup enabled by the high flux generated by a liquid jet X-ray source. This advancement holds promise in enhancing the accessibility of the technique, contrasting with the limited experiments feasible at synchrotron sources. The paper effectively elucidates the results, justifies conclusions, and exhibits a commendable scientific merit. With minor revisions, the manuscript is poised for publication in the journal.

On line 42, it is suggested to spell out the abbreviation for NPs (nanoparticles) upon its first mention.

Line 58- The X-ray source is an important aspect of X-ray imaging techniques: it has to deliver sufficient flux (photons/second) to achieve high contrast and short exposure times. – This is arguable, because flux density at the sample is more important than the flux.  

The SXM experiment description lacks a mention of the environmental conditions, specifically whether it was conducted in a gas or vacuum environment. This crucial detail should be included for a comprehensive understanding of the experimental setup.

Concerning line 212, the sentence referring to signal collection orthogonal to the pencil beam and complementarily to the transmitted photons is poorly phrased. A revised version should be crafted for clarity and precision.

For line 221, if the estimation of NP concentration has been conducted, the paper should present the numerical results. The current presentation in figures 5a-c provides only a qualitative demonstration of measured spectra, lacking the necessary information for deriving concentrations from single measurements. The text requires clarification and modification to address this discrepancy.

Reviewer 2 Report

Comments and Suggestions for Authors

The aim of the reviewed publication was to present the methodology for examining nanoparticles located in the organic matter of living cells in their native environment and to image the distribution of nanoparticles containing selected transition metals in the internal organs of living mice. The research used a plasma source of soft X-rays for imaging organic matter (soft X-ray microscope SXM) and an X-ray tube with a liquid anode for imaging the decomposition of transition metals (X-ray fluorescence imaging (XFI). The publication is extremely valuable and innovative. In the research the use of laboratory X-ray sources instead of sources based on synchrotron radiation was proposed. However, the work requires significant additions. The authors should report the spatial resolving power they achieved in both SXM and XFI. The XFI method was used to image transition metals Mo and Ru in the internal organs of mice, i.e. organs with sizes smaller than about 1 cm. Due to the methodological nature of the research, the authors should specify which elements with smaller atomic numbers than Ru and Mo can be tested. Absorption of the excited characteristic radiation in tissues on the way from the site of excitation to the skin surface imposes a limitation in terms of o elements emitting characteristic radiation with energies lower than those of the Kα Mo and Ru lines. Regarding the XFI method, the scanning step and accumulation time of the excited intensities for the pixel should be specified. More data should be presented regarding the SDD detector, i.e. what was the thickness of the detector's active layer that determined its K line detection efficiency for MO and Ru and its energy resolving power. Regarding the text of the publication, it might be better to place the % Materials and Methods chapter before chapter 2 Results and Discussion.

Comments on the Quality of English Language

 Minor editing of English language required

Reviewer 3 Report

Comments and Suggestions for Authors

Please see document attached

Comments on the Quality of English Language

Minor editing of English language required

Round 2

Reviewer 2 Report

Comments and Suggestions for Authors I believe that the authors followed the comments presented in the review and
improved and significantly supplemented the text of the publication. After the
additions, the manuscript is fully suitable for publication.

Author Response

I appreciate your thoughtful review of the manuscript. Your input has significantly improved the manuscript, and it's reassuring to know that, in your view, it is now ready for publication. Thanks again for your consideration and valuable feedback.

Reviewer 3 Report

Comments and Suggestions for Authors

Comments on the Quality of English Language

Minor editing of English language required
